# Methods to Measure Antibody Neutralization of Live Human Coronavirus OC43

**DOI:** 10.3390/v13102075

**Published:** 2021-10-14

**Authors:** Jim Boonyaratanakornkit, Anton M. Sholukh, Matthew Gray, Emily L. Bossard, Emily S. Ford, Kizzmekia S. Corbett, Lawrence Corey, Justin J. Taylor

**Affiliations:** 1Vaccine and Infectious Disease Division, Fred Hutchinson Cancer Center, Seattle, WA 98109, USA; mgray@fredhutch.org (M.G.); ebossard@fredhutch.org (E.L.B.); esford3@fredhutch.org (E.S.F.); lcorey@fredhutch.org (L.C.); jtaylor3@fredhutch.org (J.J.T.); 2Department of Medicine, University of Washington, Seattle, WA 98195, USA; 3Vaccine Research Center, National Institutes of Health, Bethesda, MD 20892, USA; kizzmekia_corbett@hsph.harvard.edu; 4Department of Laboratory Medicine and Pathology, University of Washington, Seattle, WA 98195, USA; 5Department of Immunology, University of Washington, Seattle, WA 98109, USA

**Keywords:** human coronavirus, OC43, neutralization assays, neutralizing antibodies

## Abstract

The human Betacoronavirus OC43 is a common cause of respiratory viral infections in adults and children. Lung infections with OC43 are associated with mortality, especially in hematopoietic stem cell transplant recipients. Neutralizing antibodies play a major role in protection against many respiratory viral infections, but to date a live viral neutralization assay for OC43 has not been described. We isolated a human monoclonal antibody (OC2) that binds to the spike protein of OC43 and neutralizes the live virus derived from the original isolate of OC43. We used this monoclonal antibody to develop and test the performance of two readily accessible in vitro assays for measuring antibody neutralization, one utilizing cytopathic effect and another utilizing an ELISA of infected cells. We used both methods to measure the neutralizing activity of the OC2 monoclonal antibody and of human plasma. These assays could prove useful for studying humoral responses to OC43 and cross-neutralization with other medically important betacoronaviruses.

## 1. Introduction

OC43 is one of four human coronaviruses (HCoVs) known to circulate endemically and is a major cause of upper and lower respiratory tract infections in adults and children [1]. Lower respiratory tract infection with HCoVs is associated with significant mortality in hematopoietic stem cell transplant recipients and in three recent case series OC43 was the most commonly identified of the four HCoVs in this population [2,3,4].

OC43 was first isolated in 1967 from a respiratory specimen obtained from a human adult male with a coldlike illness [5]. The viral isolate was initially cultured on human tracheal organ cultures, grown in mouse brain, and then adapted for tissue culture by passaging in primary rhesus monkey kidney cells and then BSC-1 cells [5,6,7]. OC43 belongs to the same genus Betacoronavirus as SARS-CoV-2; however, it is classified into a separate subgenus, Embecovirus. Based on phylogenetic sequence analyses, OC43 is thought to have originated from ancestral rodent-associated coronaviruses and bovine coronaviruses followed by zoonotic acquisition by humans from ungulate livestock leading to a pandemic in the late 19th century [1,8,9,10]. Since this spillover event over a century ago, the seroprevalence of OC43 in the general population is now close to 100% [11].

Neutralizing antibodies translocated from the serum into the respiratory tract or produced locally at the mucosa play a significant role in protection against many respiratory viral infections [12,13,14,15]. With the recent emergence of SARS-CoV-2, the causative agent of the COVID-19 pandemic, neutralization assays have been rapidly developed and deployed to study the antibody response and correlates of protection in animal models and humans to SARS-CoV-2 [12,16]. To identify antibodies that could cross-neutralize SARS-CoV-2 and the HCoVs, pseudotyped viral neutralization assays have been reported for OC43 [17,18]. However, the production of the pseudotyped vesicular stomatitis virus requires the simultaneous expression of bovine coronavirus hemagglutinin esterase to allow proper maturation of the OC43 spike protein. Live virus neutralization assays are considered a “gold standard” for assessing virus neutralizing potency of serum or monoclonal antibodies. To date, live virus neutralization assays using authentic OC43 have not yet been reported in the literature. We report here the development of two in vitro methods for measuring OC43 neutralizing antibodies, one using cytopathic effect and another using an ELISA of infected cells. Both methods are readily accessible with commercially available reagents, and the ELISA-based method can be scaled up into a high throughput format. We also report the isolation of an OC43 neutralizing monoclonal antibody (mAb) that can be used as a positive control in these assays.

## 2. Materials and Methods

### 2.1. Cell Lines

HCT-8 cells (ATCC, cat#CCL-244) were cultured in RPMI (Gibco, Waltham, MA, USA, cat#22400089) supplemented with 10% horse serum (Gibco, cat#26050-088) and 100 U/mL penicillin plus 100 μg/mL streptomycin (Gibco, cat#15140122). We cultured 293 F cells (Thermo Fisher, Waltham, MA, USA, cat#R79007) in Freestyle 293 media (Thermo Fisher, cat#12338026). We cultured 3T3 CD40L/IL-2/IL-21 feeder cells in DMEM supplemented with 10% fetal calf serum (Peak, Wellington, CO, USA, cat#PS-FB2), penicillin and streptomycin, plus 0.4 mg/mL geneticin as described [19]. Irradiation was performed with 5000 rads.

### 2.2. Viruses

OC43 was obtained from BEI Resources, NIAID, NIH (Bethesda, MD, USA, cat#Human Coronavirus, OC43, NR-52725). HCT-8 cells were inoculated with the OC43 stock at a MOI = 0.1 and incubated at 33 °C for 5 days. The supernatant was collected and frozen at −80 °C to generate viral working pools.

### 2.3. Human Specimens

Informed consent was obtained from all human subjects, including blood donors, involved in the study. Studies involving human spleens were deemed non-human subjects research since tissue was de-identified, otherwise discarded, and originated from deceased individuals. Tissue fragments were passed through a basket screen, centrifuged at 300× *g* for 7 min, incubated with ACK lysis buffer (Thermo Fisher, cat#A1049201) for 3.5 min, resuspended in RPMI (Gibco, cat#11875093), and passed through a stacked 500 µm and 70 µm cell strainer. Cells were resuspended in 10% dimethylsulfoxide in heat-inactivated fetal calf serum (Gibco, cat#16000044) and cryopreserved in liquid nitrogen before use. Plasma samples from SARS-CoV-2 seronegative individuals were obtained from a Fred Hutchinson Cancer Center repository assembled from a COVID-19 seroepidemiology study conducted in a single county in the western US [20]. This study was approved by the Fred Hutchinson Cancer Research Center institutional review board (#10453).

### 2.4. Immunofluorescence

HCT-8 cells were seeded into 96-well plates at a density of 20,000 cells/well two days prior to fixation with 20% methanol for 60 min at 4 °C. Wells were then blocked with 2% bovine serum albumin (Sigma, St. Louis, MO, USA, cat#A2153) in 1× Dulbecco’s phosphate-buffered saline (1× DPBS, Thermo Fisher, cat#14190250) for 30 min at room temperature. Polyclonal rabbit anti-OC43 nucleoprotein (N) antibody (SinoBiological, Beijing, China, cat# 40643-T62) was added to wells at a dilution of 1:1000 in 1× DPBS for 90 min at room temperature. Wells were washed three times with 1× DPBS. Secondary goat anti-rabbit immunoglobulin conjugated to Alexa Fluor 488 (Thermo Fisher, cat#A32731) was added to wells at a dilution of 1:1000 in 2% bovine serum albumin in 1× DPBS for 60 min at room temperature. After washing three times with 1× DPBS, images were acquired using the EVOS Cell Imaging System (Thermo Fisher).

### 2.5. Determining the Titer of OC43

HCT-8 cells were seeded in 96-well flat bottom plates and cultured for 48 h. The working pool of OC43 was diluted 1:10 and serially diluted ten-fold to a dilution of 1:10^12^ in RPMI. Fifty microliters of diluted virus were inoculated onto HCT-8 cells and allowed to adsorb for 1 h at 33 °C followed by the addition of 100 µL of RPMI. After incubating for 8 days at 33 °C, wells were examined by microscopy for cytopathic effect (CPE). The titer in fifty percent tissue culture infection dose (TCID_50_) was determined by the Reed–Muench formula [21,22].

For ELISA-based viral titration, cells were infected similarly as above at a starting dilution of 1:10^3^ and serially diluted five-fold to 9.77 × 10^9^. The plate was then incubated for 5 days at 33 °C before developed as described below in Section 2.15. The TCID_50_ was determined using the Reed–Muench formula as described [23]. A positive signal was defined as an (optical density) OD_450_ value that was equal or greater than the average OD_450_ for the “cells only” wells plus 5 times the standard deviation.

### 2.6. Expression and Purification of Spike Proteins

The design of expression plasmid for His-tagged human coronavirus spike proteins stabilized in the prefusion conformation (S2P) is previously described [24] and contains the following mutations: A1091P/L1092P for OC43, K986P/V987P for SARS-CoV-2, and N1071P/L1072P for HKU1. The expression plasmid for human parainfluenza virus type 1 (HPIV1) fusion F protein in the postfusion (postF) conformation is previously described [25]. We transfected 293 F cells at a density of 10^6^ cells/mL in Freestyle 293 media using 1 mg/mL PEI Max (Polysciences, Warrington, PA, cat#24765). Transfected cells were cultured for 7 days with gentle shaking at 37 °C. Supernatant was collected by centrifuging cultures at 2500× *g* for 30 min followed by filtration through a 0.2 µM filter. The clarified supernatant was incubated with Ni Sepharose beads overnight at 4 °C, followed by washing with wash buffer containing 50 mM Tris, 300 mM NaCl, and 8 mM imidazole. His-tagged protein was eluted with an elution buffer containing 25 mM Tris, 150 mM NaCl, and 500 mM imidazole. The purified protein was run over a 10/300 Superose 6 size exclusion column (GE Life Sciences, Issaquah, WA, cat#17–5172–01). Fractions containing the trimeric OC43 S2P or HPIV1 postF proteins were pooled and concentrated by centrifugation in an Amicon ultrafiltration unit (Millipore, Billerica, MA, cat#UFC510024) with a molecular weight cutoff of 100 kDa or 50 kDa, respectively. The concentrated sample was stored in 50% glycerol at −20 °C. Recombinant spike for NL63 (cat#40604-V08B), 229E (cat#40605-V08B), and SARS-CoV-1 (cat#40634-V08B) were obtained from Sino Biological.

### 2.7. Tetramerization of Antigens

Purified OC43 S2P and HPIV1 postF were biotinylated using an EZ-link Sulfo-NHS -LC-Biotinylation kit (Thermo Fisher, cat#A39257) using a 1:1.3 molar ratio of biotin to protein. Unconjugated biotin was removed by centrifugation using a 100 kDa or 50 kDa Amicon Ultra size exclusion column, respectively. To determine the average number of biotin molecules bound to each protein, streptavidin-R-phycoerythrin (PE) (ProZyme, Hayward, CA, cat#PJRS25) was titrated into a fixed amount of biotinylated protein at increasing concentrations and incubated at room temperature for 30 min. Samples were run on an SDS-PAGE gel (Invitrogen, Waltham, MA, USA, cat#NW04127BOX), transferred to nitrocellulose, and incubated with streptavidin–Alexa Fluor 680 (Thermo Fisher, cat#S32358) at a dilution of 1:10,000 to determine the point at which there was excess biotin available for the streptavidin–Alexa Fluor 680 reagent to bind. Biotinylated proteins were mixed with streptavidin-PE at the ratio determined above to fully saturate streptavidin and incubated for 30 min at room temperature. Unconjugated S2P was removed by centrifugation using a 300 K Nanosep centrifugal device (Pall Corporation, Port Washington, NY, USA, cat#OD300C33). PE/DyLight650 tetramers were created by mixing HPIV1 postF with streptavidin-PE pre–conjugated with DyLight650 (Thermo Fisher, cat#62279) following the manufacturer’s instructions. On average, PE/DyLight650 contained 4–8 DyLight molecules per PE. The concentration of each tetramer was calculated by measuring the absorbance of PE (578 nm, extinction coefficient = 2.0 µM^−1^ cm^−1^).

### 2.8. Tetramer Enrichment

40–80 × 10^6^ frozen spleen cells were thawed into DMEM with 10% fetal calf serum and 100 U/mL penicillin plus 100 µg/ mL streptomycin. Cells were centrifuged and resuspended 50 µL of ice-cold fluorescence-activated cell sorting (FACS) buffer composed of 1× DPBS and 1% newborn calf serum (Thermo Fisher, cat#26010074). PostF PE/DyLight650 conjugated tetramers were added at a final concentration of 25 nM in the presence of 2% rat and mouse serum (Thermo Fisher) and incubated at room temperature for 10 min. S2P PE tetramers were then added at a final concentration of 5 nM and incubated on ice for 25 min, followed by a 10 mL wash with ice-cold FACS buffer. Next, 50 μL of anti-PE-conjugated microbeads (Miltenyi Biotec, Auburn, CA, USA, cat#130-090-855) were added and incubated on ice for 30 min, after which 3 mL of FACS buffer was added and the mixture was passed over a magnetized LS column (Miltenyi Biotec, cat#130-042-401). The column was washed once with 5 mL ice-cold FACS buffer and then removed from the magnetic field and 5 mL ice and 5 mL ice-cold FACS buffer was forced through the unmagnetized column twice using a plunger to expel cells retained in the column as the bound cell fraction.

### 2.9. Flow Cytometry

Cells were incubated in 50 μL of FACS buffer containing a cocktail of antibodies for 30 min on ice prior to washing and analysis on a FACS Aria (BD). Antibodies included anti-IgM FITC (G20-127, BD), anti-CD19 BUV395 (SJ25C1, BD), anti-CD3 BV711 (UCHT1, BD), anti-CD14 BV711 (M0P-9, BD), anti-CD16 BV711 (3G8, BD), anti-CD20 BUV737 (2H7, BD), anti-IgD BV605 (IA6-2, BD), and a fixable viability dye (Tonbo Biosciences, cat#13-0870-T500). B cells were individually sorted into flatbottom 96-well plates containing feeder cells that had been seeded at a density of 28,600 cells/well one day prior in 100 µL of IMDM media (Gibco, cat#31980030) containing 10% fetal calf serum, 100 U/mL penicillin plus 100 µg/mL streptomycin, and 2.5 µg/mL amphotericin. These 3T3 feeder cells produce CD40L, IL2, and IL21 to stimulate antibody secretion into culture supernatants [19]. B cells sorted onto feeder cells were cultured at 37 °C for 13 days.

### 2.10. Neutralization Screen

For neutralization screening of culture supernatants, HCT-8 cells were seeded in 96-well flat bottom plates and cultured for 48 h. After 13 days of culture, 40 µL of B cell culture supernatant was mixed with 25 µL of the OC43 working pool diluted to 50 TCID_50_/well for 1 h at 33 °C. HCT-8 cells were then incubated with 50 µL of the supernatant/virus mixture for 1 h at 33 °C to allow viral adsorption. Next, each well was overlaid with 100 µL RPMI. Wells were examined at 8 days postinfection for CPE.

### 2.11. B Cell Receptor Sequencing and Cloning

For individual B cells sorted onto feeder cells, supernatant was removed after 13 days of culture, plates were immediately frozen on dry ice, stored at −80 °C, thawed, and RNA was extracted using the RNeasy Micro Kit (Qiagen, Germantown, MD, cat#74034). The entire eluate from the RNA extraction was used in the reverse transcription and two rounds of PCR, as previously described [26]. Sequences were analyzed using IMGT/V-Quest to identify V, D, and J gene segments. Paired heavy chain VDJ and light chain VJ sequences were cloned into pTT3-derived expression vectors containing the human IgG1, Igκ, or Igλ constant regions using In-Fusion cloning (Clontech, Mountain View, CA, USA, cat#638911) as previously described [26,27].

### 2.12. Monoclonal Antibody Production

Secretory IgG was produced by co-transfecting 293 F cells at a density of 10^6^ cells/mL with the paired heavy and light chain expression plasmids at a ratio of 1:1 in Freestyle 293 media using 1 mg/mL PEI Max. Transfected cells were cultured for 7 days with gentle shaking at 37 °C. Supernatant was collected by centrifuging cultures at 2500× *g* for 15 min followed by filtration through a 0.2 µM filter. Clarified supernatants were then incubated with Protein A agarose (Thermo Fisher, cat#22812) followed by washing with IgG binding buffer (Thermo Fisher, cat#21007). Antibodies were eluted with IgG Elution Buffer (Thermo Fisher, cat#21004) into a neutralization buffer containing 1 M Tris-base, pH 9.0. Purified antibody was concentrated and buffer exchanged into 1× DPBS using an Amicon ultrafiltration unit with a 50 kDa molecular weight cutoff.

### 2.13. Biolayer Interferometry

Biolayer interferometry (BLI) assays were performed on the Octet. Red instrument (ForteBio, Menlo Park, CA, USA) at room temperature with shaking at 500 rpm. Anti-human IgG capture sensors (ForteBio, cat#18–5060) were loaded in kinetics buffer (1× DPBS with 0.01% bovine serum albumin, 0.02% Tween 20, and 0.005% NaN3, pH 7.4) containing 40 µg/mL purified mAb for 150 s. After loading, the baseline signal was recorded for 60 s in kinetics buffer. The sensors were then immersed in kinetics buffer containing 1 µM purified OC43 S2P for a 300 s association step followed by immersion in kinetics buffer for an additional 600 s dissociation phase. Curve fitting was performed using a 1:1 binding model and ForteBio data analysis software.

### 2.14. CPE-Based OC43 Neutralization Assay

HCT-8 cells were seeded in 96-well flat bottom plates and cultured for 48 h. The OC2 mAb was diluted at a starting concentration of 123,000 ng/mL and serially diluted four-fold to 0.5 ng/mL in RPMI. Diluted OC2 mAb was mixed with 50-fold the TCID_50_ of OC43 for 1 h at 33 °C. Fifty µL of this mixture was then inoculated onto HCT-8 cells, and virus was allowed to adsorb for 1 h at 33 °C followed by the addition of 100 µL of RPMI. After incubating for 8 days at 33 °C, wells were examined by microscopy for CPE.

### 2.15. ELISA-Based OC43 Microneutralization Assay

HCT-8 cells were seeded at 20,000 cells/well in flat-bottom 96-well plates and cultivated in RPMI containing 10% horse serum and penicillin-streptomycin at 37 °C for 2 days, reaching near confluency. Serially diluted antibody was mixed with fifty-fold of TCID_50_ of OC43 per well in serum-free RPMI and incubated for 1 h at 33 °C. The virus/antibody mixtures were then transferred onto HCT-8 cells and incubated at 33 °C in CO_2_ incubator. At day 5 postinfection, media were aspirated and cells were fixed with either 70% MeOH or 85% acetone for 15 min, both prechilled to −20 °C. Plates were rinsed with 1× DPBS and blocked with 2.5% Blotting grade blocker (Bio-Rad, cat#1706404) and 0.05% Tween-20 (Sigma, cat#P9416) in 1× DPBS for 1 h at 37 °C. After blocking, plates were washed once with PBS-T (1× DPBS, 0.05% Tween-20) and then incubated with rabbit polyclonal anti-OC43 N antibody (SinoBiological, cat#40643-T62) or rabbit polyclonal anti-OC43 spike (S) antibody (MyBioSource, San Diego, CA, USA, cat#MBS1493076) for 1 h at room temperature. Plates were washed three times with PBS-T and then with horseradish peroxidase-conjugated goat anti-rabbit IgG (Jackson ImmunoResearch, West Grove, PA, USA, cat#111-035-144) for 1 h at room temperature. The assay was developed with One Step Ultra TMB (Thermo Fisher, cat#34028), and the reaction was stopped with 2 N sulfuric acid (Fisher Scientific, Waltham, MA, USA, cat#NC0301071). OD at 450 and 620 nm were captured with a SpectraMax M2 microplate reader (Molecular Devices, San Jose, CA, USA). Neutralization was defined as the antibody concentration or reciprocal of plasma dilution that reduced OD relative to virus control wells (cells + virus only) after subtraction of background OD in the cells-only control wells. The 50% neutralizing dilution (ND_50_) for plasma samples was defined as the reciprocal of the plasma dilution at which the OD was reduced by 50% compared to virus control wells after subtraction of background OD.

### 2.16. Luminex OC43-Binding IgG Assay

The assay was performed as described [28]. Briefly, serial plasma dilutions were incubated with MagPlex beads conjugated with OC43 S2P followed by incubation with anti-human IgG Fc-PE (Southern Biotech, Birmingham, AL, USA, cat#9040-09). The background signal was established by measuring the mean fluorescence intensity (MFI) of beads conjugated to antigens incubated in assay buffer instead of plasma and was subtracted from all readings. The concentration of antigen-specific IgG was estimated using a standard curve based on the measurement of MFI for serial dilutions of standard IgG (Sigma, cat#I4506) captured by MagPlex beads conjugated with anti-human IgG Fab-specific (Southern Biotech, cat#2041-01). MFI readings and their associated IgG concentrations were fitted to a four-parameter logistic curve (4PL) using the R packages nCal and drc.

### 2.17. Statistics

Correlations were estimated between pairs of neutralization and binding antibody readouts using Pearson’s correlation coefficient (*r*). Group means were compared using a paired two-sample *t*-test; measures in units of neutralization and IgG concentration were logged prior to estimating correlation and comparing group means. Statistical significance was based on *p* < 0.05. OD was first transformed to percent neutralization using the formula:%neut = (1 − ([OD (sample) − OD (cells only)]/[OD (virus only) − OD (cells only)])) × 100%

The neutralization vs. dilution curve was then fit with a four-parameter logistic curve (4PL) model that was used to estimate the dilution at which there would be 50% neutralization. Fitting and ND_50_ were estimated using Prism v.9.1.2 (GraphPad).

## 3. Results

### 3.1. Comparison of Methods to Determine the Titer of OC43

The technique of measuring the TCID_50_ by an end-point dilution method has been used to titer a number of viruses, especially those that do not readily form plaques. One of the oldest and simplest methods relies on direct examination for CPE by light microscopy. Monolayers of HCT-8 cells infected with OC43 displayed cytopathic effect by day 8 postinfection (Figure 1a). Infected cells appeared rounded and contained cytoplasmic vacuoles. Indirect immunofluorescence using an anti-OC43 N antibody confirmed the presence of OC43 infection (Figure 1b). A working pool of OC43 was made by harvesting supernatant from HCT-8 infected cells on day 8 postinfection. Using CPE and the Reed–Muench method, the TCID_50_ of this working pool was 5.0 × 10^5^ TCID_50_/mL (Figure 1c).

To detect OC43 infection using an ELISA-based approach, we first tested the effect of different fixatives. Although acetone fixation yielded a higher signal above background compared to methanol fixation in a dose-dependent manner with anti-OC43 S antibody, the OD was overall significantly lower using a polyclonal anti-OC43 S antibody as compared to a polyclonal anti-OC43 N antibody (Figure 1d). There was no significant difference in the OD when acetone was used as a fixative instead of methanol with anti-OC43 N antibody (Figure 1d). We also noted that acetone-fixed monolayers were more prone to slipping off the plates during washes, whereas methanol-fixed monolayers were more strongly attached. The end-point dilution assay employing an anti-OC43 N antibody and methanol fixation yielded a titer of 7.1 × 10^5^ TCID_50_/mL (Figure 1e). Therefore, we chose to use methanol fixation and anti-OC43 N antibody in the ELISA of infected cells in subsequent experiments.

### 3.2. Identification of a B Cell Producing an OC43 Neutralizing Antibody

In order to develop and test a quantitative neutralization assay, we sought to first isolate an OC43 neutralizing mAb that could be used as a positive control and could also aid in assay standardization between different laboratories. Since virtually all adults have been exposed to OC43, we did not need to screen donors for prior infection [11]. We biotinylated OC43 S2P and mixed it with PE-labeled streptavidin to create an OC43 S2P tetramer. We then enriched for OC43 S2P-binding B cells from the spleen of a human donor using magnetic microbeads conjugated to antibodies targeting PE (Figure 2a). We included a “decoy” tetramer consisting of biotinylated HPIV1 postF and streptavidin conjugated to PE/DyLight650 to allow for identification and exclusion of B cells binding non-specifically and B cells binding to PE or streptavidin [26,29,30,31,32]. We sorted individual isotype-switched, decoy negative, OC43 S2P-binding B cells onto irradiated 3T3 feeder cells expressing CD40L, IL-2, and IL-21 in 96-well plates, as described [19,26]. We screened culture supernatants qualitatively for neutralization of OC43 by mixing 40 µL of supernatant with 50 TCID_50_ of OC43 and assessing each well for CPE on day 8 postinfection. We sequenced the immunoglobulin genes from one of the isotype-switched, OC43 S2P-binding B cells which had supernatant that prevented CPE (Figure 2a, red box). The B-cell receptor had an IgA isotype and utilized the heavy chain variable allele 4–34 and the kappa light chain variable allele 1–39 (Figure 2b). The variable heavy and light chain genes of this class-switched B-cell receptor were cloned and expressed as an IgG1 which we named OC2. We confirmed binding of OC2 to OC43 S2P by biolayer interferometry. Under avid binding conditions, OC2 bound with high apparent affinity to OC43 S2P (Figure 2c). OC2 did not bind to the spike proteins of other coronaviruses (Figure 2d).

### 3.3. Comparison of Methods to Determine the Neutralizing Titer of an OC43 Neutralizing Monoclonal Antibody

Using the OC2 mAb, we compared in vitro microneutralization assays with inhibition of CPE or OD reduction in an ELISA of OC43-infected HCT-8 cells. The lowest concentration of OC2 tested that prevented CPE was 30.8 µg/mL (Figure 3a). CPE was observed in all wells at the next lower concentration of 7.7 µg/mL. Using the ELISA of infected cells with anti-OC43 N antibody, the 50% inhibitory concentration (IC_50_) for OC2 was determined to be 4.9 µg/mL (Figure 3b). Therefore, neutralization of OC43 by the OC2 mAb was confirmed and could be quantified by microneutralization assays using both inhibition of CPE and reduction in OD on an ELISA of infected cells. We also tested another OC43-binding mAb in both microneutralization assays and it appeared non-neutralizing [33]. Thus, both assays successfully allowed discrimination between neutralizing and non-neutralizing antibodies.

### 3.4. Measurement of Neutralizing Antibodies in Human Serum

With validation of the positive control OC2 mAb, we next applied the ELISA-based microneutralization assay to serial dilutions of plasma from human donors. Dose-dependent neutralization of OC43 was observed for all samples tested (Figure 4a). The WHO international standard for anti-SARS-CoV-2 antibody, a pool of convalescent plasma from donors with high levels of anti-SARS-CoV-2 antibodies [34], was also included in the assay and was found to have a high level of neutralizing activity against OC43. The ND_50_ was significantly correlated with total levels of IgG binding to OC43 S in the same plasma samples (*N* = 28) as measured by a Luminex binding antibody assay (Pearson *r* = 0.55, 95% confidence interval 0.21–0.77, *p* = 0.003) (Figure 4b). Therefore, a microneutralization assay using an ELISA of infected cells can be applied to plasma samples, and neutralizing titers are correlated with concentrations of OC43 S-binding IgG.

## 4. Discussion

We have developed in vitro methods to titer replication-competent OC43 and for measuring the neutralizing activity of antibodies against OC43. One titration assay assessed CPE and another used an ELISA of infected cells with a commercially available polyclonal anti-OC43 N antibody. Both methods yielded similar viral titers, indicating the robustness of both approaches. Although the commercially available polyclonal anti-OC43 S antibody we tested in the ELISA-based method did not yield a strong signal above background, it is possible other preparations of polyclonal anti-OC43 S antibodies or monoclonal antibodies could perform better.

While OC43 pseudotyped viral neutralization assays have recently been reported [17,18], to our knowledge, this study represents the first description of an in vitro live virus neutralization assay for OC43. We used CPE- and ELISA-based methods to measure neutralization activity against live virus. Neutralization can also be measured by plaque reduction or by flow-based assays, as have been performed for other respiratory viruses, such as respiratory syncytial virus and human metapneumovirus [35,36]. A flow-based method was not tested here but could provide another quantitative measure of neutralization.

In order to develop and optimize an in vitro neutralization assay with live virus, we isolated a novel mAb called OC2 from a human donor that binds with high apparent affinity to OC43 S2P. While others have isolated mAbs from a SARS-CoV-2 convalescent individual or from mice immunized with MERS-CoV that cross-neutralize OC43 in a pseudotyped virus assay, OC2 is the first human mAb, to our knowledge, that specifically binds and neutralizes OC43 [17,18]. We used OC2 to compare measurements of neutralizing activity based on inhibition of CPE or OD reduction in an ELISA of infected cells. Since the method using CPE relies on complete inhibition of CPE, it is not surprising that the neutralizing titer determined by this was approximately 6-fold greater than of that determined by ELISA, which identifies the concentration of antibody at which viral detection is reduced by 50%. The in vitro potency of OC2 is in the µg/mL range and thus is likely too low to be beneficial clinically. However, OC2 could serve not only as a useful positive control but also as a standard for sensitive quantification of binding and neutralization assays given that it is an easily renewable and scalable source. It also could be used for assay harmonization and standardization across different laboratories.

The ELISA-based microneutralization assay can be easily scaled up as a high throughput tool to identify more potent neutralizing OC43 mAbs and to study antibody responses in human specimens, including serum, nasal washes, and bronchoalveolar lavages. One limitation of the current study was that we did not analyze OC43-negative sera that could provide a better understanding of assay specificity and sensitivity. Given the close to 100% seroprevalence of OC43 [11], it is nearly impossible to obtain such samples, except from children after waning of maternal antibodies but before OC43 infection. Instead, as a negative control, we used an OC43-binding antibody that did not neutralize the virus, thus confirming the specificity of both assays [33]. We also found that all 25 plasma samples tested had some level of OC43 neutralizing activity that positively correlated with the concentration of OC43-binding IgG. Another interesting observation from using the microneutralization assay was the high OC43 neutralizing activity observed in the first WHO international standard for SARS-CoV-2 immunoglobulin. The WHO standard is pooled convalescent plasma from individuals with high titers of antibody against SARS-CoV-2 [34]. Further studies will be needed to determine if SARS-CoV-2 infection is associated with a rise in OC43 neutralizing antibodies and if this phenomenon could be related to cross-reactivity to the spike proteins of both viruses.

## 5. Conclusions

CPE-based and ELISA-based methods have been developed to titer OC43 and determine neutralizing activity of mAbs and plasma against live virus. In addition, we report the discovery of an OC43 neutralizing mAb named OC2 which may be useful as a positive control and standard in OC43 neutralization assays.

## Figures and Tables

**Figure 1 viruses-13-02075-f001:**
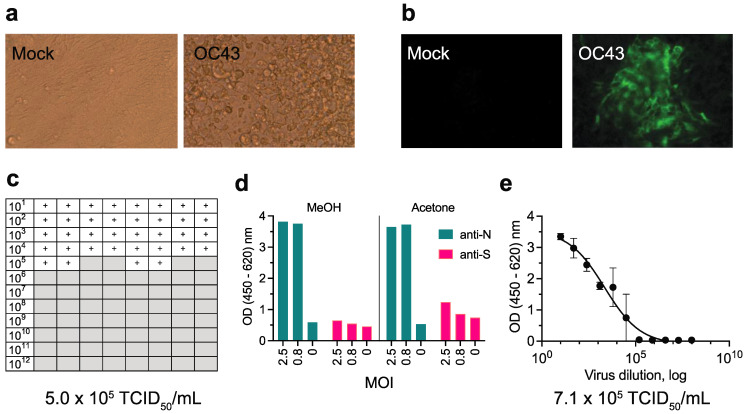
Comparison of methods to determine the titer of OC43. HCT-8 monolayers were infected with serial 10-fold dilutions of OC43 viral stocks from 1:10 to 1:10^12^. (**a**,**c**) Wells were visually examined for cytopathic effect (CPE) with light microscopy and scored as having CPE (+) or not having CPE (gray boxes) at day 8 postinfection. (**b**) Virus was also visualized by fluorescent microscopy using rabbit anti-OC43 N antibody followed by AF488-conjugated goat anti-rabbit antibody on day 3 postinfection. “Mock” refers to negative control wells mock-infected with PBS. (**c**) The TCID_50_ was calculated based on CPE on day 8 using the Reed–Muench method. Each box represents a well of a 96-well microtiter plate. The titer was calculated from two independent experiments, each run in quadruplicate. (**d**) Comparison of using methanol (MeOH) versus acetone fixation and using rabbit anti-OC43 N antibody versus rabbit anti-OC43 S antibody to detect OC43 infection of HCT-8 cells. The secondary antibody used was horseradish peroxidase-conjugated goat anti-rabbit antibody and the OD was measured on day 5 postinfection with OC43 at a multiplicity of infection (MOI) of 2, 0.8, and 0 (no virus). (**e**) The TCID_50_ was calculated based on staining with rabbit anti-OC43 N antibody followed by horseradish peroxidase-conjugated goat anti-rabbit antibody on day 5 postinfection. Each data point represents the mean and standard error of four independent replicates. The ODs obtained from the “cells only” negative controls were defined as background and subtracted from all measurements.

**Figure 2 viruses-13-02075-f002:**
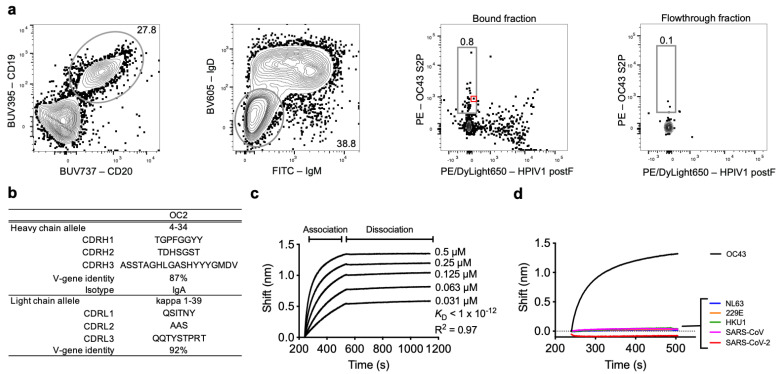
Identification of an OC43 neutralizing B cell by flow cytometry. Spleen from an adult donor was enriched for cells that could bind to OC43 S2P. (**a**) The gating strategy to isolate live, IgD^−^/IgM^−^, OC43 S2P-binding B cells is shown. Numbers indicate the percentage in the gate of total cells shown in the plot. The immunoglobulin genes were sequenced and cloned from the B cell indicated by the red square to produce the neutralizing mAb OC2. (**b**) Allele usage, CDR sequences, and isotype of OC2. (**c**) Binding kinetics of OC2 with OC43 S2P, ranging from 0.5 µM to 0.031 µM. (**d**) Binding during the association phase of OC2 with 0.5 µM of OC43 S2P, SARS-CoV-2 S2P, and HKU1 and spike from NL63, 229E, and SARS-CoV. (**c**,**d**) BLI data shown is background subtracted using a negative isotype control mAb. *K*_D_ represents the apparent binding affinity measured by biolayer interferometry. R^2^ represents the coefficient of determination.

**Figure 3 viruses-13-02075-f003:**
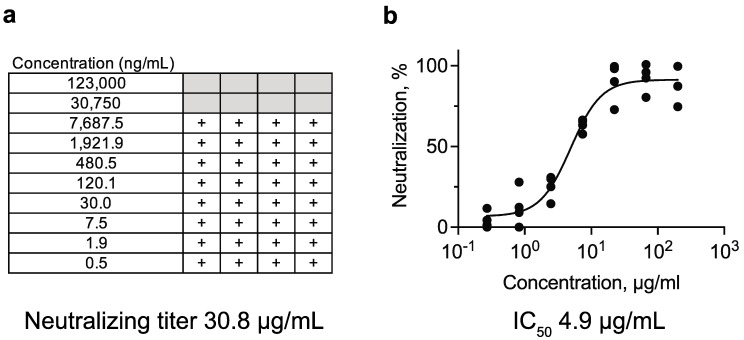
Comparison of methods to determine the neutralizing titer of an OC43 neutralizing monoclonal antibody. Serial 4-fold dilutions of the OC2 mAb were incubated with 50 TCID_50_ of OC43 for one hour and adsorbed onto monolayers of HCT-8. (**a**) Neutralization was assessed visually by the absence of CPE on day 8 postinfection. The neutralizing titer was calculated from the lowest concentration of monoclonal antibody that neutralized infectivity in at least half of parallel wells assessed on day 8. + indicates the presence of CPE. Gray boxes indicate no evidence of CPE. Each box represents a well of a 96-well microtiter plate. The titer was calculated from two independent experiments run in duplicate. (**b**) The IC_50_ was calculated based on staining with rabbit anti-OC43 N antibody followed by horseradish peroxidase-conjugated goat anti-rabbit antibody on day 5 postinfection. Each data point represents an independent replicate (*N* = 4).

**Figure 4 viruses-13-02075-f004:**
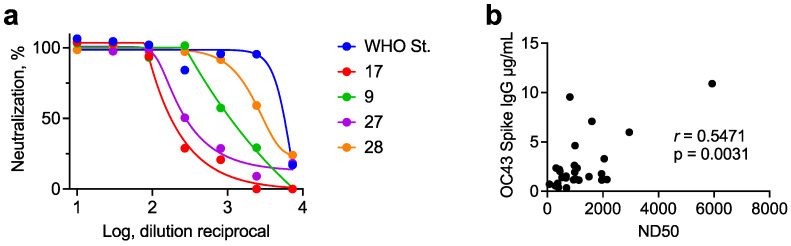
Measurement of OC43 neutralizing antibody titers in plasma using a micro-neutralization assay. (**a**) Plasma from healthy adult donors was serially diluted and incubated with 50 TCID_50_ of OC43 for one hour and adsorbed onto monolayers of HCT-8. The neutralizing titer was calculated based on staining with rabbit anti-OC43 N antibody followed by horseradish peroxidase-conjugated goat anti-rabbit antibody on day 5 postinfection. WHO St. represents the First WHO international standard for anti-SARS-CoV-2 immunoglobulin, which is a pool of convalescent plasma from donors with high levels of anti-SARS-CoV-2 antibodies [34]. (**b**) Correlation of ND50 with total levels of anti-OC43 S IgG measured by Luminex assay. *r* represents the Pearson correlation, and *p* represents the two-tailed *p* value.

## Data Availability

Not applicable.

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
