# Peer review of "Methods to Measure Antibody Neutralization of Live Human Coronavirus OC43"

_viruses, 2021, doi:10.3390/v13102075_

Round 1
Reviewer 1 Report
Authors established two methods to determine the in vitro neutralizing capacity of human antibodies against one of the most widespread human coronaviruses OC43.Authors also isolated a specific human monoclonal antibody produced by human B-Lymphocytes from human spleen. Authors found that not only the 25 plasma samples but also the spleen cells contained
antibodies against the coronavirus OC43 suggesting that cross-reaction could explain positivity against the so called new Coronavirus (SARS-CoV-2)
Comment this is a technical paper which however aims to underline the importance of "neutralizing" antibodies, which are supposed to be produced by vaccination and to prevent viral infection.
a. it has to be underlined through the text that the papers only deals with in vitro tests
b. There is no proof that serum antibodies bind to virus particles in the blood and by this means accelerate viral clearance from the blood.
c. the presence of antibodies in the nasal mucosa of previously infected patients or animals could lead to the inefficient penetration of the mucoasal cells by the virus. This has however never been studied.
d. the papers mentioned in the literature (e.g.16) do not help to solve the problem. This paper even invite to be cautious while interpreting experiments performed in animal models.
e. most of the experiments lack clear negative and positive controls (or are not mentioned)
f. this reviewer believes that informed consent from organ and blood sample donors or from their relatives should be obtained bevor materials can be used.
Reviewer 2 Report
The article describes the generation of a new antibody (OC2) specific to OC43 and the establishment of a CPE- and a cell-ELISA based neutralization assay. The description of the assays and the discussion of the results are done well. However, the following few points need to be dealt with:
Tables and axis labels in Figures (Figure 1c and Figure 3) must be checked for typing errors. Good documentation of the assay establishment and validation is crucial for such an article. Mistyping should not occur to this extent (conversion to PDF?).
It is mentioned that OC2 binds to OC43 S2P: has the OC2 antibody been tested for cross reactivities to other viral proteins of OC43 or other coronaviruses? Would it be possible to use it also for neutralization assays for animal betacoronaviruses? It has been mentioned in line 394 that OC2 specifically neutralizes OC43, how has this been shown?
For the generation of the monoclonal antibody only in vitro systems have been used. It is good to see that this animal-free method of generating humanized monoclonal antibodies has been performed successfully.
Round 2
Reviewer 1 Report
Authors made changes according to the suggestions of the reviewer